# GAD-VLP: Geometric Adversarial Detection for Vision-Language Pre-Trained Models

## Abstract

Vision-language pre-trained models (VLPs) have been deployed in numerous real-world applications; however, these models are vulnerable to adversarial attacks. Existing adversarial detection methods have shown their efficacy in single-modality settings (either vision or language), while their performance on VLPs, as multimodal models, remains uncertain. In this work, we propose a novel aspect of adversarial detection called GAD-VLP, which detects adversarial examples by exploiting vision and joint vision-language embeddings within VLP architectures. We leverage the geometry of the embedding space and demonstrate the unique characteristics of adversarial regions within these models. We explore the embedding space of the vision modality or the combined vision-language modalities, depending on the type of VLP, to identify adversarial examples. Some of the geometric methods do not require explicit knowledge of the adversary's targets in downstream tasks (e.g., zero-shot classification or image-text retrieval), offering a model-agnostic detection framework applicable across VLPs. Despite its simplicity, we demonstrate that these methods deliver a nearly perfect detection rate on state-of-the-art adversarial attacks against VLPs, including both separate and combined attacks on the vision and joint modalities.

## 1 Introduction

Vision-language pre-trained models (VLPs) enable the interpretation of both visual and textual data by learning joint representations of multimodal inputs. This makes them highly effective for tasks requiring a deep understanding of both images and text. VLPs have achieved state-of-the-art results in many multimodal tasks (Yin et al., 2023a; Xu et al., 2023; Gandhi et al., 2023), including image-text retrieval (Chen et al., 2020a), visual question answering (Lu et al., 2019), and zero-shot classification (Radford et al., 2021). Despite their success, VLPs remain vulnerable to adversarial examples (Zhang et al., 2022a; Schlarmann & Hein, 2023), posing a challenge to their robustness in real-world safety-critical applications. The safety of vision-language models is vital in critical tasks like report generation and visual question answering, especially in healthcare, where errors can lead to misdiagnosis or inappropriate treatment. Inaccuracies in image-text retrieval may also have serious consequences in high-stakes domains, making the robustness of these models essential.

Recent works have explored adversarial training as a means to improve the zero-shot robustness of VLPs (Mao et al., 2022; Wang et al., 2024; Schlarmann et al., 2024). However, adversarial training is known to be time-consuming (Madry et al., 2017; Wang et al., 2020) and often requires a trade-off between performance and robustness (Zhang et al., 2019; Tsipras et al., 2019). Detecting adversarial examples offers a more flexible alternative since it allows the model to reject queries by refusing to provide output when they are identified as adversarial.

Many methods have previously been proposed for detecting adversarial examples in unimodal models (Feinman et al., 2017; Lee et al., 2018; Ma et al., 2018; Sotgiu et al., 2020; Kherchouche et al., 2020; Aldahdooh et al., 2023). However, it remains unclear whether these methods generalize to VLPs that involve two interacting modalities. Previous work primarily focused on detecting adversarial images in classification models, which are trained using cross-entropy loss to minimize the discrepancy between predictions and labels. However, in VLPs, the training process centers around minimizing the distance between text and image embeddings. There is a lack of comprehensive investigation into detecting adversarial examples in multimodal pre-trained models, such as VLPs.

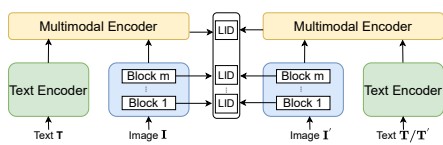 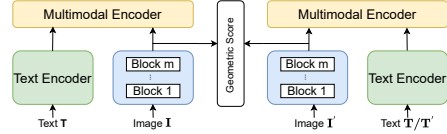

(a) Extraction of LID scores.  (b) Extraction of $k$-NN, Mahalanobis, and KDE.

Figure 1: Pipeline of geometric scores extraction for GAD-VLP.

In this paper, we propose an adversarial example detection method, GAD-VLP, which leverages geometric approaches, including local intrinsic dimensionality (LID), k-nearest neighbors distance ($k$-NN), Mahalanobis distance, and kernel density estimation (KDE), to identify adversarial images in VLPs. These metrics allow us to characterize the properties of adversarial subspaces, which is crucial for detecting adversarial examples. By evaluating the geometric properties around points, we can reveal the structural differences between adversarial and clean data regions in image classification models (Ma et al., 2018). All approaches involve extracting distances and metrics from the unimodal or multimodal encoders within VLPs, which are then used to detect adversarial examples. The overview of GAD-VLP is illustrated in Figure 1. Figure 1a presents the diagram for adversarial image detection using LID, while Figure 1b illustrates adversarial detection using the other three studied methods, $k$-NN, Mahalanobis distance, and KDE. The key difference is that LID operates in a layer-wise manner, evaluating the outputs of both multimodal layers and other intermediate layers, while the other three methods operate on the output of the image encoder.

The novelty of our research lies in exploring geometric approaches for adversarial detection in VLPs, advancing the field by providing a robust method for different types of VLPs—aligned and fused. Aligned VLPs (Zhang et al., 2022a) maintain separate embeddings for image and text modalities but ensure that they are aligned for effective interaction. In contrast, fused VLPs (Zhang et al., 2022a) integrate both image and text modalities into a single multimodal embedding. Our work reveals the unique characteristics of adversarial regions in VLPs, making them more distinguishable compared to traditional image classification models. Notably, GAD-VLP operates within the representation space, independent of specific applications, and remains effective across diverse adversarial objectives in downstream tasks, such as zero-shot classification and image-text retrieval. The main contributions of this paper can be summarized as follows:

- Adversarial Detection in VLPs: By leveraging the concept of both supervised and unsupervised geometric approaches, GAD-VLP effectively distinguishes between clean and adversarial images.

- Architecture-Independent Applicability: GAD-VLP is universally applicable across various types of VLPs. Through extensive experiments on benchmark datasets, we evaluate the methods' performance across different VLP designs, including both aligned and fused architectures. Our results demonstrate that the proposed approach is effective for adversarial detection in a wide range of VLPs, showcasing its versatility and robustness.

- Robust Detection Across Diverse Attacks: We evaluate GAD-VLP across a variety of attacks, demonstrating its effectiveness with competitive AUC scores. Our results indicate that the proposed approach maintains robust performance across different attack types, highlighting its versatility and reliability in adversarial detection.

## 2 RELATED WORKS

**Vision-Language Pre-Trained Models.** Vision-language representation learning displays superior performance across a wide range of tasks compared to visual representation learning models. For instance, CLIP uses a contrastive objective (i.e., InfoNCE loss (Oord et al., 2018)) that aims to align an image with its corresponding textual description in the feature space. VLPs aim to improve multimodal task performance by pretraining extensive image-to-text pairs (Li et al., 2022). Several recent methods utilize pre-trained object detectors with region features as a foundation for obtaining vision-language representations (Chen et al., 2020b).

There are two primary types of VLPs depending on their architectures: fused VLPs and aligned VLPs (Zhang et al., 2022a). Fused VLPs, such as ALBEF and TCL (Yang et al., 2022), utilize distinct unimodal encoders to handle token embeddings and visual characteristics separately. They subsequently employ a multimodal encoder to produce integrated multimodal embeddings by combining image and text embeddings. Conversely, aligned VLPs such as CLIP are composed solely of unimodal encoders that have separate embeddings for image and text modalities. This research specifically examines widely used architectures including both fused and aligned VLPs.

**Adversarial Robustness.** Adversarial attacks aim to deceive deep learning models into misclassifying an input (Szegedy et al., 2013). Previous works are centered around image classification. Recent studies show that vision-language models are also vulnerable to adversarial attacks. For example, Xu et al. (2018) investigated attacks on visual question-answering models by altering the image modality. In contrast, Agrawal et al. (2018); Shah et al. (2019) focused on disrupting vision-language models through text modality perturbations. Zhang et al. (2022a) explored adversarial attacks on VLPs, offering key insights into the development of multimodal attacks and improving model robustness. Building on this, Lu et al. (2023); He et al. (2023); Han et al. (2023) worked on enhancing the transferability of multimodal adversarial examples by leveraging cross-modal interactions, data augmentation, and optimal transport theory. Additionally, Yin et al. (2023b); Zhou et al. (2023) aimed to improve upon the techniques introduced by Zhang et al. (2022a). However, many attack methods are not well-suited for transformer-based VLPs and primarily target vision-language classification tasks, limiting their generalization to non-classification tasks. Therefore, we adopted the adversarial attacks presented in Zhang et al. (2022a) as our baseline.

With the rise of large-scale VLPs, their robustness towards adversarial attacks has become a major concern. For instance, Yang et al. (2021) explored the robustness of various multimodal models and proposed a defense strategy primarily designed to defend against attacks on a single modality, with unclear performance against multimodal attacks. It relies on redundancy between modalities, making it less effective when modalities provide complementary information. Fine-tuning is another popular approach to adapting pre-trained models for specific downstream tasks (Devlin, 2018). However, fine-tuning vision-language models often results in overfitting. Mao et al. (2022) addressed this issue by investigating adversarial example generation and proposing an adversarial fine-tuning algorithm guided by textual supervision. Additionally, Li et al. (2024) enhanced model robustness by utilizing a text encoder to generate fixed anchors (normalized feature embeddings) for each category and then using these anchors for adversarial training. While fine-tuning shows promising results, it suffers from issues such as inefficiency and overfitting. In light of these limitations, detecting adversarial examples presents an efficient alternative approach to defending VLPs against adversarial attacks.

**Geometric Adversarial Detection in Unimodal Models.** Several studies have employed geometric approaches to detect adversarial examples in unimodal classification models. Grosse et al. (2017) introduced the Maximum Mean Discrepancy (MMD), a kernel-based two-sample statistical test that distinguishes adversarial examples from a model's training data. This model-agnostic approach serves as a robust technique for detecting adversarial inputs. As an alternative to KDE, Ma et al. (2018) employed LID to evaluate the distance distribution of an input relative to its neighbors, capturing the local complexity of the sample's surrounding space. Additionally, Lee et al. (2018) proposed using Mahalanobis distance, leveraging Gaussian Discriminant Analysis (GDA) to detect out-of-distribution and adversarial samples through a generative classifier, offering a more refined confidence score than the traditional softmax classifier. Cohen et al. (2020) further explored $k$-NN for adversarial detection. While these methods have shown promise in unimodal settings, their efficiency in VLPs remains largely unexplored.

## 3 PRELIMINARIES

In this section, we describe the principles of adversarial attacks on VLPs and introduce the geometric approaches that are used as the basis for GAD-VLP.

### 3.1 ADVERSARIAL ATTACKS ON VLPS

All attacks in aligned VLPs are based on unimodal embeddings, as we only have access to unimodal encoders. However, in fused VLPs, two types of embeddings can be targeted: unimodal embeddings

(Uni) and multimodal embeddings (Multi). These can be further classified into full embeddings, denoted as $\text{Uni}_{\text{Full}}$ or $\text{Multi}_{\text{Full}}$, and the class embedding ([CLS]), denoted as $\text{Uni}_{\text{CLS}}$ or $\text{Multi}_{\text{CLS}}$. In this work, we focus on $\text{Uni}_{\text{CLS}}$ image attacks for CLIP, and both $\text{Uni}_{\text{CLS}}$ and $\text{Multi}_{\text{CLS}}$ attacks for ALBEF and TCL. The [CLS] embedding plays a critical role in pre-trained models, as it is used for various downstream tasks. Therefore, investigating the impact of attacks on the [CLS] embedding in VLPs is important. However, the [CLS] concept does not apply directly to CLIP models, as they can use either a ViT or CNN for image encoding. For CLIP with ViT, we treat the [CLS] embedding explicitly, while for the CNN variant, we consider the final embedding analogous to the [CLS] embedding for consistency in the remainder of the paper. For simplicity, we will refer to unimodal attacks as $\text{Sep}_{\text{uni}}$ and multimodal attacks as $\text{Sep}_{\text{multi}}$, omitting further use of the [CLS] notation.

Here, we introduce two baseline adversarial attacks—Sep-Attack and Co-Attack—based on the framework by Zhang et al. (2022a). Sep-Attack perturbs the image and text modalities separately, maximizing the adversarial perturbation using Kullback–Leibler (KL) divergence loss for the embedding-wise representation. For text perturbations, the method constrains the perturbation to a specific number of tokens based on the BERT attack. In contrast, Co-Attack jointly targets both modalities, shifting the targeted embedding away from the original. This attack applies to both fused and aligned VLPs, with separate perturbations calculated for unimodal and multimodal embeddings. Further technical details and mathematical formulations of Sep-Attack and Co-Attack can be found in Appendix A.1.

## 3.2 Geometric Approaches

**Local Intrinsic Dimension (LID)**   LID is a concept of dimensionality modeling (Karger & Ruhl, 2002; Houle et al., 2012) that quantifies the intrinsic dimensionality in the proximity of a specific point of interest in the dataset. LID evaluates the rate at which the number of encountered data objects grows as the distance from the point increases (Houle, 2013). It is a statistical model that expands upon the generalized expansion dimension model and presupposes the presence of an unknown smooth distribution of distances from a reference point (Houle, 2017). LID focuses on estimating the intrinsic dimensionality within a localized region surrounding a data point.

In practice, this quantity needs to be estimated with the query point $x$ and a set of reference points that can be used to select its nearest neighbors (Levina & Bickel, 2004; Amsaleg et al., 2015). For a given reference sample $x$ drawn from the data distribution $P$, the maximum likelihood estimator of LID is defined as follows:

$$\hat{\text{LID}}_d(x) = \left(-\frac{1}{k}\sum_{i=1}^{k}\log\frac{r_i(x)}{r_{\max}(x)}\right)^{-1}. \tag{1}$$

In this context, $r_i(x)$ represents the distance between $x$ and its $i$-th closest neighbor within a sample of $k$ points taken from $P$, and $r_{\max}(x)$ refers to the greatest distance between neighbors. In this work, we refer 'LID' as the quantity of $\hat{\text{LID}}_d(.)$.

**Mahalanobis Distance**   The Mahalanobis distance (McLachlan, 1999) measures the distance between data points in a way that accounts for the covariance structure of the data, making it a useful tool for evaluating the similarity between different data points. Unlike the Euclidean distance, the Mahalanobis distance incorporates feature correlations and is scale-invariant. For a $p$-dimensional data point $x = (x_1, x_2, ..., x_p)^T$ with mean vector $\mu = (\mu_1, \mu_2, ..., \mu_p)$ and covariance matrix $\Sigma$, the Mahalanobis distance between $x$ and the distribution characterized by $\mu$ and $\Sigma$ is given by:

$$D_M(x) = \sqrt{(x-\mu)^T\Sigma^{-1}(x-\mu)}. \tag{2}$$

This formulation captures the deviation of $x$ from the mean, highlighting how its position differs from the distribution of the data. A larger Mahalanobis distance indicates that $x$ is more likely to be an outlier or come from a different distribution.

**Kernel Density Estimation**   Kernel density estimation (KDE) is a non-parametric technique used to estimate the probability density function of a dataset without assuming a parametric form for the underlying distribution. This method estimates the density of a point population in an arbitrary $N$-dimensional space using a finite sample (Botev et al., 2010). For a random multivariate sample

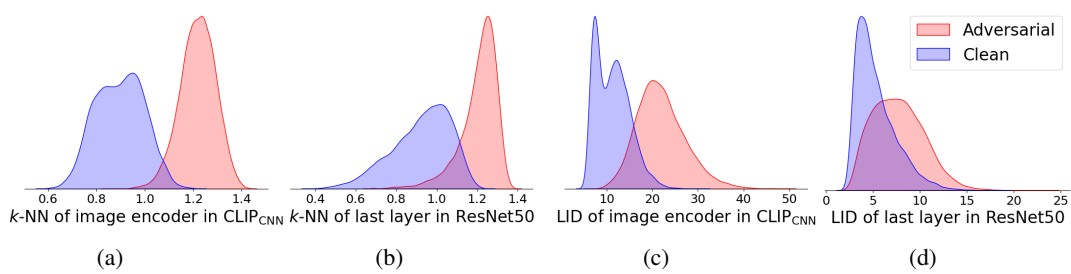

(a)      (b)      (c)      (d)

Figure 2: Comparison of the $k$-NN and LID distributions between the CLIP$_{\text{CNN}}$ image encoder and the ResNet50 using ImageNet data.

$x_i = (x_{i1}, x_{i2}, ..., x_{in})^T$ drawn from a distribution with density $f$, the kernel density estimate is defined as:

$$\hat{f}(y; H) = \frac{1}{n} \sum_{i=1}^{n} K_H(y, x_i), \tag{3}$$

where $K_H(\cdot, \cdot)$ denotes the kernel function and $H$ is the bandwidth matrix, which is both symmetric and positive definite. The kernel function $K_H$ is often chosen to be Gaussian with bandwidth $\sigma$, given by $K_\sigma(x, X_i) \sim \exp\left(-\frac{\|x - X_i\|^2}{\sigma^2}\right)$. The bandwidth matrix $H$ adjusts the kernel's shape and smoothing across dimensions, allowing for effective density estimation in multivariate contexts.

## 4 DETECTING ADVERSARIAL SAMPLES USING GEOMETRIC APPROACHES IN VLPS

In this section, we motivate the use of geometric methods. We illustrate their effectiveness through an example and compare geometric distances between VLPs and traditional classifiers. Finally, we detail the proposed framework, GAD-VLP.

### 4.1 SENSITIVITY AS A MOTIVATION FOR GENERALIZATION OF GEOMETRIC METHODS TO VLPS

In multimodal models, adversarial images are generated by maximizing the KL divergence loss between clean and perturbed embeddings (Zhang et al., 2022a). This attack method aims to produce perturbations that shift the perturbed samples away from the distribution of the original clean data, causing the input to deviate from its natural distribution, as shown in Figure 3. As a result, the perturbed input exhibits distinct characteristics that no longer align with the clean distribution.

VLPs integrate both visual and textual information, allowing them to capture complex relationships between modalities and represent a wider range of features in the data. When adversarial attacks are applied, the introduced perturbations exploit this complexity, resulting in adversarial points that diverge significantly from clean samples. We hypothesize that these adversarial examples occupy a higher-dimensional space in VLPs than in traditional models, reflecting their enhanced representation of multimodal interactions that may not be fully captured by traditional architectures.

To support this hypothesis, we analyze the $k$-NN and LID distributions of image embeddings in CLIP$_{\text{CNN}}$ with ResNet50 as the image encoder, comparing them to those in the traditional ResNet50 model. In this context, the architecture of the CLIP$_{\text{CNN}}$ image encoder aligns with that of the traditional model. It can be observed in Figure 2 that both $k$-NN distance and LID are able to separate samples from clean and adversarial data. Our findings reveal that LID values are higher in the CLIP$_{\text{CNN}}$ model compared

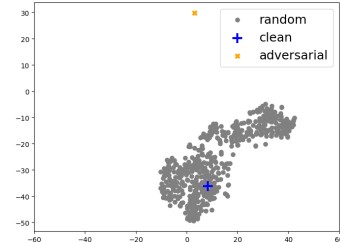

Figure 3: t-SNE visualization of CLIP$_{\text{CNN}}$ image features. Gray represents randomly selected data from CIFAR10, blue represents a clean data point, and orange represents the adversarial counterpart of the blue sample.

to the traditional ResNet50 model, suggesting that adversarial points exist in a more complex, higher-dimensional space compared to the ResNet50 model trained with a supervised objective under the same attack. Furthermore, while the range of $k$-NN distances is similar across both models, we notice that the distinction between clean and adversarial $k$-NN values is more pronounced in the CLIP$_{\text{CNN}}$ model, which improves adversarial detection. This greater separation allows adversarial points to be more clearly distinguished from clean embeddings, facilitating their identification.

In addition to this empirical analysis, Amsaleg et al. (2020) demonstrated that higher dimensionality exhibits greater sensitivity to adversarial perturbations. Using Theorem 2 of (Amsaleg et al., 2020), for a fixed choice of the ratio $\frac{k_t}{k_x}$, where $k_t$ is a target expected rank, while $k_x$ is the expected rank within the distribution of $x$, it can be shown that the proportion of perturbation required to achieve the target rank decreases as the LID increases. Specifically, for large LID values, the amount of perturbation required scales as (Amsaleg et al., 2020):

$$\delta > \frac{1}{\text{LID}(x)} \ln(\frac{k_t}{k_x}) + \epsilon + o(\frac{1}{\text{LID}(x)}). \tag{4}$$

Equation 4 indicates that as LID increases, the perturbation needed to change the rank in the neighborhood of a point decreases. Since the CLIP$_{\text{CNN}}$ model has larger LID values compared to standard classifiers, the same amount of perturbation results in more significant changes in the ranks within the neighborhood of points in the latent space. This heightened sensitivity to perturbations facilitates the detection of adversarial points in the CLIP$_{\text{CNN}}$ model compared to a traditional classifier. Thus, the larger LID values in CLIP$_{\text{CNN}}$ facilitate more effective adversarial detection.

## 4.2 GAD-VLP

GAD-VLP comprises three primary steps: generation, extraction, and detection. Following prior work, we assume that the defender has access to a subset of the data and that the initial dataset $D_c$ is free of adversarial examples. All initial samples $\{(x_i^j, x_t^j)\}_{j=1}^N$, where $x_i$ denotes the image input and $x_t$ denotes the text input, are clean. We denote the clean samples as $\{(x_i^j, x_t^j)\}_{j=1}^N \in D_c$, and the adversarial subset as $\{(x_i'^j, x^j)\}_{j=1}^N \in D_a$. In the case where both modalities are perturbed, the adversarial set is represented as $\{(x_i'^j, x_t'^j)\}_{j=1}^N \in D_a$. The defender aims to accurately detect adversarial samples, particularly those where the image is perturbed. Adversarial image detection is framed as a binary classification problem, distinguishing between adversarial and clean samples.

In the first step of the process, generation, adversarial examples are created from clean samples using adversarial attacks. We generated adversarial attacks based on the entire dataset, resulting in a balanced distribution of adversarial and clean samples for both testing and evaluation, specifically comprising equal proportions of each type. For a detailed description of the adversarial examples generation process, refer to Appendix A.1

In the extraction step, we begin by extracting clean unimodal embeddings, $z_i = E_i(x_i)$, and multimodal embeddings, $z_m = E_m(x_i, x_t)$, where $E_i$ refers to the image encoder and $E_m$ to the multimodal encoder. Next, we extract adversarial embeddings, represented as $z_i' = E_i(x_i')$ and $z_m' = E_m(x_i', x_t')$. Then, the geometric scores for both clean and adversarial images are computed using the extracted embeddings.

For Mahalanobis distance and KDE, we utilize $z_i$ from the clean training data, extracting the mean vector ($\mu$) and covariance matrix ($\Sigma$), and KDE functions ($\hat{f}(z_i; H)$), where $H$ is the bandwidth matrix. The Mahalanobis distance and KDE scores are then computed for both clean and adversarial test data, with respect to ($\mu, \Sigma$) for Mahalanobis distance and $\hat{f}(z_i; H)$ for KDE. For $k$-NN, the $k$-NN distances of embeddings are calculated by using the clean embeddings as the reference.

For LID, we adopt a layer-wise extraction approach following Ma et al. (2018). In fused VLPs, we also include the LID of the multimodal encoder $z_m$ as an additional feature alongside the layers of the image encoder, which improves detection performance against multimodal attacks. The complete procedure for computing these values for adversarial image detection is outlined in Algorithm 1 in Appendix A.2. $S_{(N,l)}$ and $S'_{(N,l)}$ represents the extracted clean and adversarial scores, where $l = 1$ for $k$-NN, Mahalanobis, and KDE, and $l$ denotes the number of layers for LID. These scores serve as the prepared features for the detection phase.

To enhance computational efficiency, we employ minibatch sampling for the extraction of LID and $k$-NN scores, particularly for large datasets. While processing the entire dataset is possible, it is often prohibitively expensive. Previous studies (Ma et al., 2018) demonstrate that minibatch sampling can be used to approximate the local neighborhood characteristics.

In the final stage, adversarial image detection is formulated as a binary classification problem, distinguishing between adversarial and clean samples. To make this distinction, we define a function $g(\cdot)$, which determines whether an image is perturbed. Adversarial features $S'$ are labeled as one, while clean features $S$ are labeled as zero. The dataset is then divided into training and testing subsets. For $k$-NN, Mahalanobis, and KDE, threshold-based detection is applied according to Equation 5 where $t$ represents the threshold, while for LID, the extracted features are used to train a binary classification model. The details on training the binary classification using LID values are provided in Appendix A.2.

$$f(x_i, x_t) = \begin{cases} 1 & \text{if } s_i > t, \\ 0 & \text{if } s_i \le t, \end{cases} \tag{5}$$

Notably, the specific task for which the VLPs' head was originally trained—classification or retrieval—is not central to our approach. We leverage only the embeddings from the image encoder and, when applicable, the multimodal encoder's embeddings, allowing our approaches to be flexibly applied across various downstream tasks. Both Mahalanobis distance and KDE methods are widely used for adversarial detection due to their capacity to model clean data distributions and identify outliers. However, they may require labeled data or strong distributional assumptions, limiting their flexibility. In contrast, $k$-NN and LID offer label-free advantages, making them suitable for unsupervised tasks common in VLPs. This flexibility is particularly valuable in datasets that lack explicit labels but include captions, allowing $k$-NN and LID to detect adversarial examples and assess perturbations effectively, even without labeled data.

## 5 EXPERIMENTS

In this section, we demonstrate the effectiveness of GAD-VLP in distinguishing adversarial images within VLPs. We evaluate four types of geometric methods, including LID, $k$-NN distance, Mahalanobis distance, and KDE for zero-shot classification, and LID and $k$-NN for image-retrieval tasks. We utilize the MCM (Ming et al., 2022) method as a baseline that is used in the concept of VLPs to compare the geometric approaches.

### 5.1 DATASETS AND MODEL

**Datasets.** We evaluate zero-shot classification with ImageNet (Deng et al., 2009), CIFAR10, CIFAR100 (Krizhevsky et al., 2009), STL-10 (Coates et al., 2011), and Food-101 (Bossard et al., 2014). For classification datasets, we use the text prompts (Radford et al., 2021) for the model with the pattern of "a photo of a $c$", where $c$ is the name of the class. We evaluate image-text retrieval on commonly used datasets, including Flickr30K (Young et al., 2014) and MS-COCO (Lin et al., 2014).

**Model.** We assess two well-known types of VLPs: aligned and fused VLPs. For the aligned VLPs, we evaluate CLIP (Radford et al., 2021), with two image encoders: $\text{CLIP}_{\text{ViT}}$ (using ViT-B/16) and $\text{CLIP}_{\text{CNN}}$ (using ResNet50). For the fused VLPs, we examine ALBEF (Li et al., 2021) and TCL (Yang et al., 2022). ALBEF and TCL contain an image encoder, a text encoder, and a multimodal encoder. These models use a 12-layer ViT-B/16 (Dosovitskiy et al., 2020) as the image encoder and initialize it with weights pre-trained on ImageNet-1k from (Deng et al., 2009). An input image $I$ is transformed into a series of embeddings: $\{v_{\text{cls}}, v_1, \ldots, v_N\}$, where $v_{\text{cls}}$ corresponds to the embedding of the [CLS] token. Both the text and multimodal encoders utilize a 6-layer transformer (Vaswani, 2017). The text encoder is initialized with the first 6 layers of the $\text{BERT}_{\text{base}}$ (Devlin, 2018) model, while the multimodal encoder is initialized with the final 6 layers of $\text{BERT}_{\text{base}}$. The text encoder processes input text $T$ into a sequence of embeddings $\{w_{\text{cls}}, w_1, \ldots, w_N\}$, which are subsequently passed to the multimodal encoder. Image features are combined with the text embeddings via cross-attention at every layer of the multimodal encoder.

**Metric and Adversarial Attack.** For assessment, we employ the following metrics: (1) the false positive rate (FPR), and (2) the area under the receiver operating characteristic curve (AUC). We

Table 1: A comparison of the discrimination power (AUC score) among MCM and GAD-VLP framework using LID, $k$-NN, Mahalanobis (denoted as Mah.) and KDE in an aligned VLP, $\text{CLIP}_{\text{CNN}}$, and a fused VLP, ALBEF.

| Model | Method | Attack | CIFAR10 | | CIFAR100 | | ImageNet1k | | STL10 | | Food101 | |
|---|---|---|---|---|---|---|---|---|---|---|---|---|
| | | | AUC | FPR95 | AUC | FPR95 | AUC | FPR95 | AUC | FPR95 | AUC | FPR95 |
| $\text{CLIP}_{\text{CNN}}$ | MCM | $\text{Sep}_{\text{uni}}$ | 65.47 | 82.88 | 41.13 | 94.15 | 86.10 | 60.35 | 95.82 | 17.92 | 91.70 | 40.18 |
| | | Co-Attack | 67.10 | 79.54 | 43.99 | 93.21 | 80.83 | 68.38 | 94.10 | 25.64 | 82.14 | 64.38 |
| | LID | $\text{Sep}_{\text{uni}}$ | 100 | 0.00 | 100 | 0.00 | 99.31 | 1.87 | 100 | 0.00 | 99.98 | 0.06 |
| | | Co-Attack | 100 | 0.00 | 100 | 0.00 | 99.50 | 1.62 | 100 | 0.00 | 99.95 | 0.08 |
| | $k$-NN | $\text{Sep}_{\text{uni}}$ | 100 | 0.00 | 100 | 0.00 | 99.65 | 1.62 | 100 | 0.00 | 100 | 0.00 |
| | | Co-Attack | 100 | 0.00 | 100 | 0.00 | 99.67 | 0.89 | 100 | 0.00 | 100 | 0.00 |
| | Mah. | $\text{Sep}_{\text{uni}}$ | 100 | 0.00 | 100 | 0.00 | 96.62 | 9.32 | 99.88 | 0.33 | 99.79 | 1.16 |
| | | Co-Attack | 100 | 0.00 | 100 | 0.00 | 97.28 | 7.97 | 99.80 | 0.59 | 99.38 | 2.32 |
| | KDE | $\text{Sep}_{\text{uni}}$ | 100 | 0.00 | 100 | 0.00 | 98.72 | 7.24 | 99.87 | 0.26 | 100 | 0.00 |
| | | Co-Attack | 100 | 0.00 | 100 | 0.00 | 99.33 | 2.81 | 99.85 | 0.33 | 100 | 0.00 |
| ALBEF | MCM | $\text{Sep}_{\text{uni}}$ | 91.20 | 29.02 | 82.80 | 49.19 | 92.15 | 25.38 | 96.83 | 16.02 | 90.26 | 37.03 |
| | | $\text{Sep}_{\text{multi}}$ | 47.43 | 98.23 | 33.55 | 99.56 | 63.03 | 97.59 | 65.32 | 86.60 | 41.98 | 99.46 |
| | | Co-Attack | 93.34 | 24.64 | 82.67 | 47.27 | 92.14 | 24.94 | 96.45 | 19.70 | 81.29 | 74.70 |
| | LID | $\text{Sep}_{\text{uni}}$ | 100 | 0.00 | 99.97 | 0.05 | 91.85 | 29.41 | 99.64 | 1.62 | 99.87 | 0.67 |
| | | $\text{Sep}_{\text{multi}}$ | 99.96 | 0.20 | 99.85 | 0.44 | 78.77 | 67.68 | 96.63 | 15.65 | 92.31 | 33.27 |
| | | Co-Attack | 100 | 0.00 | 99.98 | 0.05 | 93.85 | 20.07 | 99.85 | 0.69 | 99.92 | 0.42 |
| | $k$-NN | $\text{Sep}_{\text{uni}}$ | 100 | 0.00 | 100 | 0.00 | 98.60 | 7.23 | 99.97 | 0.19 | 99.98 | 0.04 |
| | | $\text{Sep}_{\text{multi}}$ | 99.27 | 3.05 | 99.21 | 3.25 | 51.92 | 93.61 | 75.95 | 75.75 | 86.46 | 50.96 |
| | | Co-Attack | 100 | 0.00 | 100 | 0.00 | 98.64 | 7.33 | 99.96 | 0.19 | 99.98 | 0.04 |
| | Mah. | $\text{Sep}_{\text{uni}}$ | 100 | 0.00 | 100 | 0.00 | 99.94 | 0.20 | 100 | 0.00 | 100 | 0.00 |
| | | $\text{Sep}_{\text{multi}}$ | 100 | 0.00 | 100 | 0.00 | 81.41 | 64.82 | 99.25 | 3.19 | 99.16 | 3.92 |
| | | Co-Attack | 100 | 0.00 | 100 | 0.00 | 99.93 | 0.25 | 100 | 0.00 | 100 | 0.000 |
| | KDE | $\text{Sep}_{\text{uni}}$ | 99.38 | 0.71 | 100 | 0.00 | 96.78 | 16.93 | 99.70 | 1.06 | 99.95 | 0.16 |
| | | $\text{Sep}_{\text{multi}}$ | 99.24 | 0.86 | 99.85 | 0.81 | 66.83 | 81.75 | 88.63 | 66.19 | 87.61 | 49.27 |
| | | Co-Attack | 99.38 | 0.76 | 100 | 0.00 | 96.67 | 18.09 | 99.72 | 1.00 | 99.94 | 0.16 |

follow Sep-Attack and Co-Attack methods (Zhang et al., 2022a) due to their applicability to different models and tasks. For the adversarial attack on the image modality, Sep-Attack (denoted as $\text{Sep}_{\text{uni}}$ for unimodal attack and $\text{Sep}_{\text{multi}}$ for multimodal attack, which can be done only in fused VLPs, and Co-Attack both use PGD. The maximum perturbation $\epsilon_i$ is set to $8/255$. The step size is set to $1.25$, the number of iterations is set to 10, and the maximum perturbation for text $\epsilon_t$ is set to 1 token.

**Benchmark for Comparison.** Given the limited research in adversarial detection within VLPs, we opted to include the MCM method (Ming et al., 2022) as a baseline that is used in VLPs. This method utilizes the softmax scores of the similarities between image and text embeddings in the CLIP model to detect out-of-distribution data, making it the most suitable state-of-the-art method for comparison within VLPs. We aimed to assess its effectiveness in VLPs for adversarial image detection and to compare it with GAD-VLP to demonstrate its advantage. However, the MCM method is designed for datasets with specific labels and does not apply to image-text retrieval, limiting our comparison to classification datasets. MCM returns lower values for adversarial images compared to clean images. These extracted MCM scores are then used to apply a threshold for distinguishing between adversarial and clean images.

**Settings.** Depending on the distance and metric, the following hyperparameters need to be specified: (1) the number of neighbors for estimating LID, (2) the number of neighbors for $k$-NN, (3) batch size, and (4) KDE kernel size. For the CLIP model, both the ViT-B/16 and ResNet50 image encoder architectures, we employ a batch size of 128, a $k$ value of 100 for LID, 10 nearest neighbors for $k$-NN, and KDE kernel of 0.1. For the ALBEF and TCL models, we set the batch size to 64, the $k$ value to 40 for LID, 10 nearest neighbors for $k$-NN, and KDE kernel of 0.1 for all attack scenarios.

Table 2: GAD-VLP discrimination power (AUC score) for Image-Retrieval Task with Flickr30k and COCO dataset in aligned VLPs ($CLIP_{CNN}$ and $CLIP_{ViT}$), and fused VLPs (ALBEF and TCL).

(a) Results for $CLIP_{CNN}$ and ALBEF Models

| Model | Method | Attack | Flickr30k | | COCO | |
|---|---|---|---|---|---|---|
| | | | AUC | FPR | AUC | FPR |
| $CLIP_{CNN}$ | LID | $Sep_{uni}$ | 99.67 | 1.16 | 99.78 | 0.87 |
| | | Co-Attack | 99.59 | 1.47 | 99.68 | 1.23 |
| | $k$-NN | $Sep_{uni}$ | 99.99 | 0.00 | 99.97 | 0.03 |
| | | Co-Attack | 99.95 | 0.02 | 99.83 | 0.29 |
| ALBEF | LID | $Sep_{uni}$ | 94.26 | 24.94 | 96.80 | 14.37 |
| | | $Sep_{multi}$ | 77.39 | 70.03 | 79.79 | 66.10 |
| | | Co-Attack | 94.27 | 25.01 | 96.63 | 14.54 |
| | $k$-NN | $Sep_{uni}$ | 99.00 | 3.00 | 99.45 | 3.31 |
| | | $Sep_{multi}$ | 64.68 | 87.01 | 70.70 | 75.65 |
| | | Co-Attack | 98.98 | 3.82 | 99.41 | 3.27 |

(b) Results for $CLIP_{ViT}$ and TCL Models

| Model | Method | Attack | Flickr30k | | COCO | |
|---|---|---|---|---|---|---|
| | | | AUC | FPR | AUC | FPR |
| $CLIP_{ViT}$ | LID | $Sep_{uni}$ | 99.92 | 0.23 | 99.89 | 0.31 |
| | | Co-Attack | 99.01 | 4.75 | 98.83 | 5.24 |
| | $k$-NN | $Sep_{uni}$ | 100 | 0.00 | 100 | 0.00 |
| | | Co-Attack | 99.73 | 0.44 | 99.68 | 0.92 |
| TCL | LID | $Sep_{uni}$ | 96.02 | 19.01 | 98.38 | 7.51 |
| | | $Sep_{multi}$ | 85.82 | 54.35 | 85.97 | 54.74 |
| | | Co-Attack | 96.11 | 19.05 | 98.22 | 8.65 |
| | $k$-NN | $Sep_{uni}$ | 96.58 | 18.19 | 97.73 | 9.78 |
| | | $Sep_{multi}$ | 42.19 | 93.52 | 65.09 | 83.60 |
| | | Co-Attack | 96.52 | 18.46 | 97.76 | 9.61 |

We utilized the fine-tuned image-retrieval model on the Flickr-30k dataset for both the ALBEF and TCL models across all datasets.

## 5.2 RESULTS

**Performance of GAD-VLP in Zero-Shot Classification.** As can be seen in Table 1, geometric approaches consistently outperform the MCM method across all datasets in $CLIP_{CNN}$, achieving lower FPR and higher AUC. This highlights the effectiveness of GAD-VLP in detecting adversarial samples. Notably, $k$-NN surpasses other metrics (particularly Mahalanobis and KDE) in $CLIP_{CNN}$, with LID showing comparable performance to $k$-NN in this context.

While recent advancements in VLPs have largely centered on CLIP, we extended our evaluation to include the ALBEF model. ALBEF, which features a fused multimodal encoder alongside separate encoders for image and text, presents a different dynamic: Mahalanobis distance outperforms other metrics (especially KDE), demonstrating its strength in detecting adversarial examples in the ALBEF model. Additionally, in the context of multimodal attacks ($Sep_{Multi}$), LID shows similar performance, underscoring the importance of multimodal embeddings in detecting such attacks.

For $CLIP_{CNN}$, $k$-NN proves particularly effective due to the detailed representation of visual inputs provided by the image embeddings from the encoder, allowing $k$-NN to effectively capture local differences between clean and adversarial samples. Since adversarial perturbations in CLIP typically result in subtle shifts within the embedding space, $k$-NN's neighbor-based distance calculations are well-suited to identifying outliers. Similarly, LID, by capturing the local dimensionality of the space, further emphasizes its strength in detecting adversarial samples in $CLIP_{CNN}$.

In ALBEF, the Mahalanobis distance, applied to image embeddings, excels in identifying deviations from the expected distribution. Although ALBEF integrates multimodal information, adversarial perturbations within the image modality still produce measurable changes in the embedding space. Mahalanobis, with its capacity to model the covariance structure of clean image embeddings, effectively identifies these deviations without relying heavily on multimodal interactions. Furthermore, LID, when incorporating multimodal embedding outputs as a feature in the detection process ($Sep_{Multi}$), demonstrates performance comparable to Mahalanobis distance. Additional results for $CLIP_{ViT}$ and TCL, provided in Appendix A.3, show consistent patterns with those in this subsection.

**Performance of GAD-VLP in Image-Text Retrieval.** We also aimed to demonstrate that the detection of adversarial images in VLPs is not constrained to classification tasks with datasets having specific labels. For this purpose, we evaluated the performance of the image-text retrieval task on two datasets, to assess if the method applies to non-classification datasets, Flickr30k and COCO. Due to the lack of labels in this task, we only examine the LID and $k$-NN distance since they do not require labels. The results can be seen in Table 2. The performance of the $CLIP_{CNN}$, $CLIP_{ViT}$, ALBEF and, TCL models is comparable to their performance on classification datasets. For both the

Table 3: Generalization of GAD-VLP to $Sep_{uni}$ with different baseline attacks in $CLIP_{ViT}$ for STL10. The reported results are AUC.

| Attack | Method | | | |
|--------|--------|------|--------|------|
|        | LID    | $k$-NN | Mahal. | KDE  |
| PGD    | 99.99  | 100  | 99.99  | 99.97 |
| FGSM   | 91.93  | 98.99 | 99.59 | 99.12 |
| R-FGSM | 75.39  | 74.12 | 94.11 | 83.91 |
| I-FGSM | 99.99  | 100  | 99.99  | 99.96 |
| MI-FGSM | 99.97 | 100  | 99.99  | 99.96 |

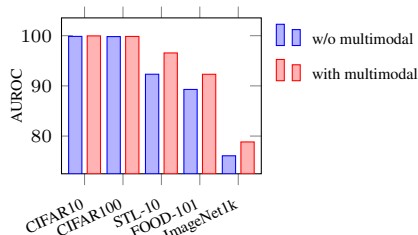

Figure 4: The effect of the multimodal encoder of the LID detection in multimodal-based attacks on fused VLPs.

COCO and Flickr30k datasets, each image is annotated with five captions. To maintain consistency, As Co-Attack requires a matching prompt to simultaneously attack both the image and the associated text, we select the first caption as the target text for the Co-Attack method.

### 5.3 ABLATION STUDY

**The Effect of Multimodal Embeddings on Adversarial Detection.** We evaluate the impact of including a multimodal layer on LID value computation. As shown in Figure 4, the exclusion of this feature results in a reduction in AUC scores in most datasets. This indicates that for attacks based on the multimodal encoder, the inclusion of this layer is crucial. The LID difference in this layer is significantly more noticeable compared to other layers.

**Generalization to Different Gradient-based Attacks.** It is important to investigate whether the detector can effectively detect samples from different attack strategies. To address this, we conduct an evaluation to assess its ability to generalize to new attack baselines beyond the PGD-based attacks. Specifically, for LID method, we train the detector using PGD-based attacks and then evaluate its performance on samples generated from other attack strategies, including FGSM (Goodfellow et al., 2014), R-FGSM (Tramèr et al., 2018), I-FGSM (Kurakin et al., 2018), and MI-FGSM (Dong et al., 2018). Both the training and test datasets follow the same preparation method as in our previous experiments, with PGD-based attacks applied to the training set, while evaluated attacks are used for the test set. For the other methods ($k$-NN, Mahalanobis, and KDE), since they rely on thresholds, we simply assess their performance against the new attack types. The results, presented in Table 3, demonstrate that the geometric-based method shows significant generalizability across various gradient-based attack strategies.

### 6 CONCLUSION

In this paper, we address, for the first time, the problem of detecting adversarial attacks against VLPs, which are increasingly applied across diverse domains. We demonstrate that our framework, GAD-VLP, which utilizes simple geometric metrics applied to image or joint representations, can effectively detect adversarial examples. Our detection approach generalizes across various tasks and state-of-the-art VLPs—$CLIP_{CNN}$, $CLIP_{ViT}$, ALBEF, and TCL. A key insight from our study is the increased separation between clean and adversarial geometric scores in the latent space of the CLIP model, in contrast to traditional classifiers. This distinction enhances the effectiveness of geometric scores for adversarial detection. Our results demonstrate that the proposed framework performs robustly across different VLP architectures, whether aligned or fused, and is effective against state-of-the-art adversarial attacks. Moreover, the detection process is independent of the downstream tasks. An open issue for future research is the examination of text-exclusive attacks, a prominent concern within VLPs. Further exploration is necessary to identify robust methodologies for leveraging the embeddings in the detection of adversarial attacks in the text domain.

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

# A  APPENDIX

## A.1  ATTACKS ON VLPS

In this part, we provide an extended explanation of the adversarial attacks discussed in Section 3.1. We provide additional technical details and mathematical formulations for the Sep-Attack and Co-Attack methods, including how they perturb multimodal and unimodal embeddings in vision-language models. These formulations expand upon the description of the attacks provided in the main text, offering a deeper dive into their mechanisms and objectives.

**Sep-attack**  Sep-Attack (Zhang et al., 2022a) was introduced to perturb image and text modalities separately. As VLPs can be used for non-classification tasks without explicit labels, they propose using Kullback–Leibler (KL) divergence loss instead of commonly used cross entropy. In this way, the Sep-Attack aims to maximize the KL divergence loss ($\mathcal{L}$) of the embedding-wise representation to produce an adversarial perturbation:

$$\delta_i = \epsilon_i . sign(\nabla_{x_i'}\mathcal{L}(E_i(x_i'), E_i(x_i))). \tag{6}$$

For perturbing the text modality, the text perturbation can be denoted as follows:

$$\delta_t = \arg\max_{x_t^i}(\left\|E_t(x_t') - E_t(x_t)\right\|) - x_t. \tag{7}$$

Maximum perturbation $\epsilon_t$ is constrained in the way that how many tokens are perturbed in each prompt based on BERT attack (Li et al., 2020). For the attack on multimodal embedding, the unimodal encoder is replaced with the multimodal encoder, which is denoted as $E_m(\cdot, \cdot)$. It is worth mentioning that this attack is only applicable to fused VLPs like ALBEF, which have multimodal encoders. The attack on the image is as follows:

$$\delta_i = \epsilon_i . sign(\nabla_{x_i'}\mathcal{L}(E_m(E_i(x_i'), E_t(x_t)), E_m(E_i(x_i), E_t(x_t)))). \tag{8}$$

**Co-Attack**  In Sep-attack, combining attacks on the text and image may be less effective than attacking them individually. To overcome this challenge, Co-Attack (Zhang et al., 2022a) was developed to jointly target the image modality and the text modality. It aims to shift the perturbed multimodal embedding away from the original embedding or the perturbed image-modal embedding away from the perturbed text-modal embedding. The versatility of Co-Attack allows it to be applied to both fused VLPs and aligned VLPs, making it suitable for attacking both multimodal and unimodal embeddings. The attack on unimodal embedding aims to find the perturbation $\delta_i$ that satisfies:

$$\arg\max_{\delta_i}\mathcal{L}(E_i(x_i'), E_t(x_t)) + \alpha_1\mathcal{L}(E_i(x_i'), E_t(x_t')). \tag{9}$$

The attack on multimodal embedding is as follows:

$$\arg\max_{\delta_i}\mathcal{L}(E_m(E_i(x_i'), E_t(x_t')), E_m(E_i(x_i), E_t(x_t')))$$
$$+\alpha_2\mathcal{L}(E_m(E_i(x_i'), E_t(x_t')), E_m(E_i(x_i), E_t(x_t))). \tag{10}$$

$\alpha_1$ and $\alpha_2$ are hyper-parameters that control the contributions of the second term.

## A.2  ALGORITHM AND DETAILS

The details of the GAD-VLP are presented in Algorithm 1. Specifically, line 2 outlines the generation step discussed in Section 4.2, while lines 4 to 13 are related to the extraction steps. Lines 16 to 18 detail the detection process. For training the LID detection model, we used the extracted $S_{(N,l)}$, where $N$ is the number of samples and $l$ is the number of layers from which features are extracted. We then divided the extracted scores into two parts: training and testing. The training data was used as features to train a binary classification model.

For the CLIP$_{\text{CNN}}$ with the CIFAR-10 dataset, feature extraction for detection methods such as MCM, KDE, Mahalanobis, and $k$-NN takes less than 2 minutes, while the LID-based method requires about 9 minutes on an NVIDIA H100 GPU. Detailed time costs are provided in Table 4. Our framework is

| Method | LID | $k$-NN | Mahalanobis | KDE | MCM |
|--------|-----|--------|-------------|-----|-----|
| **Score** | 546.11 | 103.19 | 33.08 | 69.82 | 98.48 |

Table 4: Comparison of computational cost (seconds) for different methods of GAD-VLP framework on CIFAR-10 with CLIP$_{\text{CNN}}$.

significantly more efficient compared to re-training or fine-tuning CLIP for robustness. For example, linear-probe CLIP (Radford et al., 2021) takes approximately 13 minutes, CoOp (Zhou et al., 2022) requires 14 hours and 40 minutes, and CLIP-Adapter (Peng et al., 2021) takes about 50 minutes on a n a single NVIDIA GeForce RTX 3090 GPU (Zhang et al., 2022b).

---

**Algorithm 1** GAD-VLP

**Input**: A pre-trained model, consisted of a image encoder $E_i(.)$, a text encoder $E_t(.)$, and in the case of being fused $E_m(.)$, Clean data $D_c = (x_i, x_t)_{i=1}^N$, and $L = [l_1, l_2, ..., l_f]$ selected layers for embedding extraction

1: **for** $j = 1$ **to** $length(D_c)$ **do**
2:    $X' = Attack(X)$
3:    **for** $l \in L$ **do**
4:       $z_{(j,l)} = E_i^l(x_i)$
5:       $z'_{(j,l)} = E_i^l(x'_i)$
6:       $s_{lid(j,l)} = LID(z_{clean(j,l)})$
7:       $s_{lid'(j,l)} = LID(z_{adv(j,l)})$
8:    **end for**
9:    $s_{knn(j)} = k\text{-NN}(z_{(j,l_f)})$ , $s_{mah.(j)} = \text{Mah.}(z_{(j,l_f)})$ , $s_{kde(j)} = \text{KDE}(z_{(j,l_f)})$
10:   $s'_{knn(j)} = k\text{-NN}(z'_{(j,l_f)})$ , $s'_{mah.(j)} = \text{Mah.}(z'_{(j,l_f)})$ , $s'_{kde(j)} = \text{KDE}(z'_{(j,l_f)})$
11:   **if** Attack is based on Multimodal Embedding **then**
12:      $s_{lid(j,m+1)} = LID(E_m(x_i, x_t))$
13:      $s_{lid'(j,m+1)} = LID(E_m(x'_i, x'_t))$
14:   **end if**
15: **end for**
16: $Y_{neg} = [0]_N, Y_{pos} = [1]_N, Y = [Y_{neg}, Y_{pos}]$
17: $X = [S, S']$
18: Detection Model for $(X, Y)$

---

### A.3 EXTENDED EVALUATION

Table 5 presents the results of adversarial detection for zero-shot classification in the CLIP$_{\text{ViT}}$ and TCL models. The findings are consistent with those shown in Table 1.

### A.4 SENSITIVITY TO LOCALITY

Adversarial detection methods based on local analysis, such as $k$-NN and LID, rely on the locality hyperparameter $k$ to define the neighborhood size, which can have a substantial impact on their detection performance. To explore the sensitivity of these methods to the choice of $k$ in detecting adversarial examples within VLPs, we varied $k$ across the values [10, 20, 30, 40, 50] for the adversarial detection for Sep$_{\text{uni}}$ attack in CLIP$_{\text{ViT}}$. As shown in Figure 5, our results reveal that $k$-NN demonstrates greater stability in adversarial detection compared to LID, maintaining consistent performance across different values of $k$. This highlights the robustness of $k$-NN when applied in the context of adversarial detection, whereas LID appears more sensitive to changes in $k$.

Table 5: A comparison of the discrimination power (AUC score) among MCM and GAD-VLP framework using LID, $k$-NN, Mahalanobis (denoted as Mah.) and KDE in an aligned VLP, CLIP$_{\text{ViT}}$, and a fused VLPs, TCL.

| Model | Method | Attack | CIFAR10 | | CIFAR100 | | ImageNet1k | | STL10 | | Food101 | |
|---|---|---|---|---|---|---|---|---|---|---|---|---|
| | | | AUC | FPR95 | AUC | FPR95 | AUC | FPR95 | AUC | FPR95 | AUC | FPR95 |
| CLIP$_{\text{ViT}}$ | MCM | Sep$_{\text{uni}}$ | 76.47 | 88.24 | 72.09 | 67.83 | 86.06 | 54.55 | 94.84 | 26.79 | 94.51 | 25.32 |
| | | Co-Attack | 80.21 | 73.54 | 68.37 | 76.24 | 84.14 | 58.93 | 95.51 | 20.73 | 89.78 | 40.95 |
| | LID | Sep$_{\text{uni}}$ | 100 | 0.00 | 100 | 0.00 | 99.23 | 4.57 | 99.99 | 0.00 | 99.98 | 0.02 |
| | | Co-Attack | 100 | 0.00 | 100 | 0.00 | 97.09 | 15.30 | 99.74 | 0.64 | 99.54 | 1.49 |
| | $k$-NN | Sep$_{\text{uni}}$ | 100 | 0.00 | 100 | 0.00 | 99.98 | 0.00 | 100 | 0.00 | 100 | 0.00 |
| | | Co-Attack | 100 | 0.00 | 100 | 0.00 | 98.67 | 6.64 | 99.99 | 0.00 | 100 | 0.00 |
| | Mah. | Sep$_{\text{uni}}$ | 100 | 0.00 | 100 | 0.00 | 99.85 | 0.94 | 99.99 | 0.06 | 99.98 | 0.14 |
| | | Co-Attack | 100 | 0.00 | 100 | 0.00 | 99.18 | 3.07 | 99.97 | 0.06 | 99.82 | 0.82 |
| | KDE | Sep$_{\text{uni}}$ | 100 | 0.0 | 100 | 0.00 | 99.95 | 0.10 | 99.97 | 0.19 | 100 | 0.00 |
| | | Co-Attack | 100 | 0.00 | 100 | 0.00 | 98.79 | 6.35 | 99.82 | 0.39 | 100 | 0.0 |
| TCL | MCM | Sep$_{\text{uni}}$ | 76.91 | 55.63 | 62.15 | 73.78 | 90.49 | 32.02 | 94.82 | 18.45 | 76.76 | 71.88 |
| | | Sep$_{\text{multi}}$ | 46.63 | 99.06 | 37.01 | 97.24 | 64.85 | 87.65 | 73.34 | 77.49 | 46.94 | 95.84 |
| | | Co-Attack | 80.82 | 45.65 | 69.05 | 68.32 | 92.74 | 26.71 | 97.13 | 13.65 | 79.07 | 64.75 |
| | LID | Sep$_{\text{uni}}$ | 100 | 0.00 | 100 | 0.00 | 91.92 | 28.28 | 99.62 | 1.81 | 99.77 | 1.01 |
| | | Sep$_{\text{multi}}$ | 99.91 | 0.25 | 99.88 | 0.54 | 85.64 | 53.66 | 97.71 | 12.97 | 93.15 | 30.67 |
| | | Co-Attack | 100 | 0.00 | 100 | 0.00 | 91.02 | 30.64 | 99.89 | 0.69 | 99.79 | 0.79 |
| | $k$-NN | Sep$_{\text{uni}}$ | 100 | 0.00 | 100 | 0.00 | 93.99 | 25.77 | 99.97 | 00.06 | 99.98 | 00.06 |
| | | Sep$_{\text{multi}}$ | 99.78 | 1.28 | 99.87 | 0.59 | 32.54 | 97.98 | 85.39 | 56.04 | 92.08 | 32.53 |
| | | Co-Attack | 100 | 0.00 | 100 | 0.00 | 93.83 | 26.61 | 99.98 | 00.06 | 99.98 | 0.06 |
| | Mah. | Sep$_{\text{uni}}$ | 100 | 0.00 | 100 | 0.00 | 99.65 | 1.56 | 99.99 | 0.06 | 100 | 0.00 |
| | | Sep$_{\text{multi}}$ | 100 | 0.00 | 99.99 | 0.05 | 59.92 | 89.52 | 98.28 | 8.31 | 99.11 | 3.41 |
| | | Co-Attack | 100 | 0.00 | 100 | 0.00 | 99.64 | 1.56 | 99.99 | 0.06 | 100 | 0.00 |
| | KDE | Sep$_{\text{uni}}$ | 99.16 | 0.86 | 100 | 0.00 | 90.89 | 35.33 | 98.63 | 4.56 | 99.95 | 0.24 |
| | | Sep$_{\text{multi}}$ | 98.97 | 1.46 | 99.80 | 1.11 | 59.70 | 85.08 | 80.72 | 60.62 | 92.30 | 33.70 |
| | | Co-Attack | 99.15 | 0.86 | 100 | 0.00 | 90.98 | 36.54 | 98.62 | 4.56 | 99.96 | 0.20 |

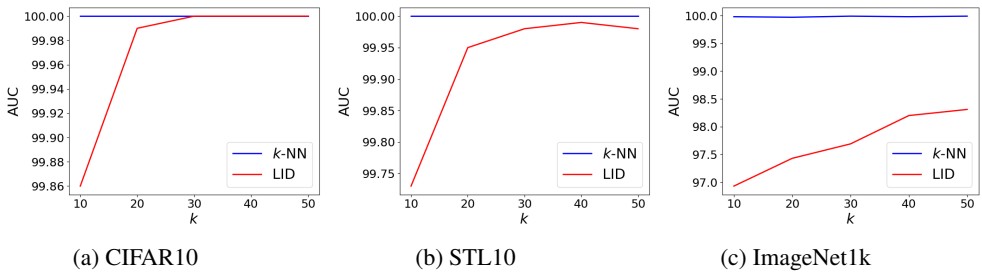

(a) CIFAR10      (b) STL10      (c) ImageNet1k

Figure 5: The detection AUC rates of local geometric approaches under varying locality k.

## A.5 EVALUATION OF ADAPTIVE ATTACKS

In this subsection, we evaluate the impact of test-time adaptive attacks on GAD-VLP. Adaptive attacks specifically target the detection mechanism by incorporating it into the optimization process for perturbation generation. Assessing their effectiveness is critical, particularly in white-box settings, where the attacker has full access to the model and can modify the optimization function to craft perturbations that directly undermine the detection method.

We have expanded our experimental setup to include adaptive attacks targeting the LID and $k$-NN detection methods. Specifically, we generate attacks designed to optimize for bypassing the detection

Table 6: GAD-VLP discrimination power (AUC score) comparison between adaptive and non-adaptive attacks for Image-Retrieval Task with Flickr30k and COCO dataset in aligned VLPs (CLIP$_{CNN}$ and CLIP$_{ViT}$), and fused VLPs (ALBEF and TCL) (Note: 'N-adaptive' refers to the Non-adaptive method.)

(a) Effect of $k$-NN adaptive Attacks

| Model | Attack | Dataset | | | |
| --- | --- | --- | --- | --- | --- |
| | | Flickr30k | | COCO | |
| | | N-adaptive | Adaptive | N-adaptive | Adaptive |
| CLIP$_{CNN}$ | Sep$_{uni}$ | 99.99 | 99.99 | 99.97 | 99.99 |
| | Co-Attack | 99.95 | 100 | 99.83 | 99.99 |
| CLIP$_{ViT}$ | Sep$_{uni}$ | 100 | 100 | 100 | 99.99 |
| | Co-Attack | 99.73 | 100 | 99.68 | 99.99 |
| ALBEF | Sep$_{uni}$ | 99.00 | 94.57 | 99.45 | 96.46 |
| | Sep$_{multi}$ | 64.68 | 87.83 | 70.70 | 92.48 |
| | Co-Attack | 98.98 | 94.50 | 99.41 | 96.45 |
| TCL | Sep$_{uni}$ | 96.58 | 94.45 | 97.73 | 94.72 |
| | Sep$_{multi}$ | 42.19 | 53.59 | 65.09 | 68.15 |
| | Co-Attack | 96.52 | 94.42 | 97.76 | 94.71 |

(b) Effect of LID adaptive Attacks

| Model | Attack | Dataset | | | |
| --- | --- | --- | --- | --- | --- |
| | | Flickr30k | | COCO | |
| | | N-adaptive | Adaptive | N-adaptive | Adaptive |
| CLIP$_{CNN}$ | Sep$_{uni}$ | 99.67 | 97.37 | 99.78 | 97.91 |
| | Co-Attack | 99.59 | 97.43 | 99.68 | 98.32 |
| CLIP$_{ViT}$ | Sep$_{uni}$ | 99.92 | 93.29 | 99.89 | 91.97 |
| | Co-Attack | 99.01 | 93.90 | 98.83 | 92.46 |
| ALBEF | Sep$_{uni}$ | 94.26 | 73.85 | 96.80 | 80.65 |
| | Sep$_{multi}$ | 77.39 | 73.66 | 79.79 | 79.43 |
| | Co-Attack | 94.27 | 73.85 | 96.63 | 79.88 |
| TCL | Sep$_{uni}$ | 96.02 | 76.90 | 98.38 | 83.53 |
| | Sep$_{multi}$ | 85.82 | 78.19 | 85.97 | 79.87 |
| | Co-Attack | 96.11 | 78.28 | 98.22 | 83.29 |

Table 7: ASR (IR@1) for the Image-Retrieval Task with Flickr30k and COCO datasets in aligned VLPs (CLIP$_{CNN}$, CLIP$_{ViT}$) and fused VLPs (ALBEF, TCL).

| Model | Attack | Flickr30k | | | COCO | | |
| --- | --- | --- | --- | --- | --- | --- | --- |
| | | Non-adaptive | LID-adaptive | $k$-NN adaptive | Non-adaptive | LID-adaptive | $k$-NN adaptive |
| CLIP$_{CNN}$ | Sep$_{uni}$ | 98.61 | 90.59 | 96.31 | 98.87 | 91.49 | 94.75 |
| | Co-Attack | 99.72 | 93.90 | 97.53 | 99.83 | 93.62 | 96.70 |
| CLIP$_{ViT}$ | Sep$_{uni}$ | 97.33 | 87.27 | 94.30 | 98.14 | 87.96 | 93.15 |
| | Co-Attack | 99.42 | 92.69 | 95.59 | 99.21 | 92.48 | 96.35 |
| ALBEF | Sep$_{uni}$ | 89.50 | 96.02 | 98.33 | 89.18 | 94.69 | 97.76 |
| | Sep$_{multi}$ | 62.57 | 58.15 | 93.55 | 44.14 | 71.79 | 95.36 |
| | Co-Attack | 93.33 | 96.43 | 98.62 | 92.35 | 97.38 | 98.41 |
| TCL | Sep$_{uni}$ | 96.18 | 96.31 | 99.54 | 97.56 | 94.61 | 99.51 |
| | Sep$_{multi}$ | 59.58 | 48.93 | 68.54 | 48.76 | 52.83 | 57.83 |
| | Co-Attack | 97.21 | 96.81 | 99.62 | 98.33 | 97.56 | 99.27 |

metrics (LID and $k$-NN) as follows:

$$L_{adaptive}(Z, Z^{'}) = L_{main}(Z, Z^{'}) + \alpha * S(Z^{'}) \tag{11}$$

Where $S(Z^{'})$ is the LID or $k$-NN function that computes the score for adversarial sample embeddings $Z^{'}$ relative to the clean sample embeddings $Z$, and we set $\alpha = 0.5$ in our experiments. The results presented in Table 6 offer insights into the resilience of the GAD-VLP framework under adaptive attacks. $k$-NN and LID are robust when detecting adaptive attacks for CLIP and reasonably effective for ALBEF and TCL. The increase in detection rates against adaptive $k$-NN attacks, particularly in the multimodal image domain, can be explained by the dynamics of attack generation. Adaptive attacks targeting the $k$-NN-based defense incorporate constraints that optimize perturbations around the $k$-NN structure. The incorporation of the multimodal encoder during attack generation modifies the data distribution, increasing the distinguishability of perturbed samples. This could cause perturbed samples to shift more significantly in the feature space, making them easier to detect.

### A.6 EVALUATION OF ATTACK SUCCESS RATES

In this subsection, we evaluated the attack success rate (ASR) specifically for image-retrieval tasks across four models and two widely used datasets. The results are shown in Tables 7 and 8.

Table 8: ASR (IR@5) for the Image-Retrieval Task with Flickr30k and COCO datasets in aligned VLPs (CLIP$_{CNN}$, CLIP$_{ViT}$) and fused VLPs (ALBEF, TCL).

| Model | Attack | Flickr30k | | | COCO | | |
|---|---|---|---|---|---|---|---|
| | | Non-adaptive | LID-adaptive | $k$-NN adaptive | Non-adaptive | LID-adaptive | $k$-NN adaptive |
| CLIP$_{CNN}$ | Sep$_{uni}$ | 97.49 | 84.45 | 93.28 | 97.33 | 83.25 | 90.46 |
| | Co-Attack | 99.30 | 89.15 | 94.87 | 99.06 | 88.83 | 93.93 |
| CLIP$_{ViT}$ | Sep$_{uni}$ | 94.60 | 79.48 | 87.54 | 95.69 | 80.12 | 88.23 |
| | Co-Attack | 98.79 | 84.85 | 91.45 | 98.92 | 86.39 | 91.61 |
| ALBEF | Sep$_{uni}$ | 85.25 | 95.37 | 96.80 | 84.05 | 94.6 | 96.11 |
| | Sep$_{multi}$ | 63.22 | 54.02 | 92.50 | 49.60 | 68.63 | 94.32 |
| | Co-Attack | 88.01 | 95.47 | 96.65 | 87.21 | 97.00 | 96.23 |
| TCL | Sep$_{uni}$ | 92.69 | 95.96 | 98.95 | 95.29 | 94.66 | 98.39 |
| | Sep$_{multi}$ | 61.31 | 45.67 | 68.68 | 54.65 | 48.68 | 60.98 |
| | Co-Attack | 93.46 | 95.92 | 98.91 | 99.49 | 97.06 | 98.44 |

