# OpenReview forum: "GAD-VLP: Geometric Adversarial Detection for Vision-Language Pre-Trained Models"
_ICLR.cc/2025/Conference — Submitted to ICLR 2025_

### Official Review · Reviewer_9b4J · 2024-10-19

**Soundness:** 4
**Presentation:** 4
**Contribution:** 2
**Rating:** 3
**Confidence:** 5

**Summary:**

This paper propose the GAD-VLP, a method for detecting adversarial attacks in VLP Models. It uses geometric metrics like Local Intrinsic Dimensionality (LID), k-Nearest Neighbors (k-NN), Mahalanobis distance, and Kernel Density Estimation (KDE) to distinguish adversarial examples. GAD-VLP works across various VLP architectures, whether they use separate or combined embeddings for vision and language inputs. The method demonstrates high detection accuracy against different attacks and is applicable to tasks like zero-shot classification and image-text retrieval. The study highlights GAD-VLP's robustness and versatility across multiple domains.

**Strengths:**

1.	The writing is clear. The formulas are correct.
2.	The experiment is multi-dimensional.
3.	The research topic is important for VLP.

**Weaknesses:**

1.	While the proposed method is effective, the metrics used are standard and not specifically tailored for the Vision-Language Pre-training (VLP) task.
2.	The paper does not sufficiently highlight the distinctions between unimodal models and VLP models, resulting in a lack of justification for the choice of metrics.
3.	Classical VLP models, such as BLIP [1] and X-VLM [2], are missing from the analysis.
4.	The adversarial attack methods utilized are limited and do not include more popular approaches like Set-level Guidance Attack (SGA) [3].

[1] BLIP: Bootstrapping Language-Image Pre-training for Unified Vision-Language Understanding and Generation

[2] Multi-Grained Vision Language Pre-Training: Aligning Texts with Visual Concepts

[3] Set-level Guidance Attack: Boosting Adversarial Transferability of Vision-Language Pre-training Models

**Questions:**

Please see the weakness.

---

> ### Author Response · Authors · 2024-11-29
> **Response to Reviewer 9b4J**
>
> Thank you for taking the time to review our paper and providing valuable feedback. Below, you will find our responses to your questions:
>
> **A1:**
>
> To the best of our knowledge, this is the first study to evaluate the detectability of adversarial images for VLPs, addressing a notable gap in the existing literature. We believe that demonstrating effective detectability using existing methods in the context of a new problem provides valuable insights to the community. Our rigorous experiments validate this novel finding, which we believe holds equal significance to proposing a new technique.
>
> ---
>
> **A2:**
>
> In Section 4.1, we specifically analyze how geometric values such as LID and k-NN behave differently in unimodal and multimodal settings, using this distinction as a key motivation for applying these metrics to VLPs.
> In unimodal models, adversarial examples typically alter embeddings in a relatively localized and lower-dimensional space. By contrast, VLPs’ multimodal nature introduces more complex interactions between vision and language embeddings, resulting in adversarial examples that occupy distinct, higher-dimensional subspaces. Our findings, supported by Figure 2, show that VLP embeddings exhibit greater separability between clean and adversarial data when compared to unimodal embeddings. For example, in CLIP, the k-NN distances of adversarial embeddings are more distinctly separated from clean embeddings than those in traditional classifiers, making these geometric metrics particularly effective for VLPs. Similarly, the sensitivity of LID to perturbations is amplified in multimodal spaces due to their inherently higher dimensionality, as discussed in Section 4.1.
>
> ---
>
> **A3:**
>
> The models we selected for evaluation—such as CLIP, ALBEF, and TCL—were chosen because they are frequently studied in the existing literature on adversarial attacks and defenses for VLPs [1-4].
>
> [1] Zhang, Jiaming, Qi Yi, and Jitao Sang. "Towards adversarial attack on vision-language pre-training models." Proceedings of the 30th ACM International Conference on Multimedia. 2022.
>
> [2] Lu, Dong, et al. "Set-level guidance attack: Boosting adversarial transferability of vision-language pre-training models." Proceedings of the IEEE/CVF International Conference on Computer Vision. 2023.
>
> [3] He, Bangyan, et al. "Sa-attack: Improving adversarial transferability of vision-language pre-training models via self-augmentation." arXiv preprint arXiv:2312.04913 (2023).
>
> [4] Han, Dongchen, et al. "Ot-attack: Enhancing adversarial transferability of vision-language models via optimal transport optimization." arXiv preprint arXiv:2312.04403 (2023).
>
> ---
>
> **A4:**
>
> SGA is designed primarily to improve the transferability of adversarial examples between different models, rather than directly increasing the ASR for a given model. Specifically, SGA enhances the ASR when adversarial examples generated for one VLP are transferred to another.

---

### Official Review · Reviewer_fo3Q · 2024-10-29

**Soundness:** 2
**Presentation:** 3
**Contribution:** 1
**Rating:** 3
**Confidence:** 3

**Summary:**

The paper looks at detection of adversarial examples for multi-modal models (CLIP, ALBEF). They extend uni-modal detection methods to vision-language based models. Experiments are conducted on classification and Image-retrieval tasks using different 'geometry of sample' based approaches to show GAD-VLP (proposed framework) works well when using different methods for detection.

**Strengths:**

- The problem being studied is important - Detecting adversarial inputs in multi-modal setup.
- The related work exploration seems sufficient.
- Diversity of evaluation tasks (classification, retrieval) and models is reasonable

**Weaknesses:**

- Overall the paper lacks technical novelty as previously used methods (MMD, KDE etc) for uni-modal detection methods are just transferred to multi-modal setup.

- The evaluation/testing setup in re to adversarial setup is not sufficient.

- The adversarial attacks do not seem strong, looking at Fig. 2 a naturally strong attack would be to add a regularizer term that enforces the image embedding (under perturbation) to stay close to the clean embedding. Testing method on such strong attacks would be nice.

- The attacks are based on10-step PGD, and versions of FGSM all at eps=8/255 (other values should have been looked at). A lot of new attacks (see the ones in [1, 2]) for CLIP like models have been proposed - testing which would have been also a valuable addition to the submission.

- For the binary classifier using LID, attacking the classifier (detector) would also be a reasonably strong attack.

[1] Schlarmann, Christian, and Matthias Hein. "On the adversarial robustness of multi-modal foundation models." Proceedings of the IEEE/CVF International Conference on Computer Vision. 2023.

[2] Mao, Chengzhi, et al. "Understanding Zero-shot Adversarial Robustness for Large-Scale Models." The Eleventh International Conference on Learning Representations.

**Questions:**

- What is the setup for retrieval is it top-1, top-5?
No other questions, some questions on choice of experiments can be inferred from Weaknesses section.

---

> ### Author Response · Authors · 2024-11-29
> **Response to Reviewer fo3Q (1/2)**
>
> Thank you for taking the time to review our paper and providing valuable feedback. Below, you will find our responses to your questions:
>
> **A1:**
>
> To the best of our knowledge, this is the first study to evaluate the detectability of adversarial images for VLPs, addressing a notable gap in the existing literature.
> We believe that demonstrating effective detectability using existing methods in the context of a new problem provides valuable insights to the community. Our rigorous experiments validate this novel finding, which we believe holds equal significance to proposing a new technique.
>
> ---
>
> **A2:**
>
> The setup for evaluation of the attack success rate is based on the top-1, and top-5 for image-retrieval tasks. We believe this is standard for this task. We appreciate it if the reviewer could make a more specific suggestion regarding alternatives.
>
> ---
>
> **A3:**
>
> Thank you for your helpful suggestion. To address this, we have expanded our experimental setup to include adaptive attacks targeting the LID and $k$-NN detection methods. Specifically, we generate attacks designed to optimize for bypassing the detection metrics (LID and k-NN) as follows:
>
> $L_{adaptive}(Z,Z^{'}) = L_{main}(Z,Z^{'}) + \alpha * S(Z^{'})$
>
> Where $S(Z^{'})$ is the function LID or $k$-NN function that computes the score for adversarial embeddings $Z^{'}$ with reference to the clean embeddings $Z$, we put $\alpha= 0.5$ in our experiments. KNN  and LID are both robust when detecting adaptive attacks for CLIP and reasonably effective for ALBEF and TCL. The increase in detection rates against adaptive k-NN attacks can be explained by the dynamics of attack generation. Adaptive attacks targeting the k-NN-based defense incorporate constraints that optimize perturbations around the k-NN structure. The incorporation of the multimodal encoder during attack generation modifies the data distribution, increasing the distinguishability of perturbed samples. This could cause perturbed samples to shift more significantly in the feature space, making them easier to detect. The results are presented in the table below, with additional datasets detailed in subtables a and b of Table 6 in the updated draft (Section A.5 in the Appendix).
>
> | Model            | Attack          | Non-adaptive | Adaptive |
> |-------------------|-----------------|-------------------------|--------------------------|
> | CLIP$\_{\text{CNN}}\$ | Uni            | 99.99                  | 99.99                   |
> |                  | Co             | 99.95                  | 100                  |
> | CLIP$\_{\text{ViT}}\$ | Uni            | 100                 | 100                  |
> |                  | Co             | 99.73                  | 100                  |
> | ALBEF            | Uni            | 99.00                  | 94.57                   |
> |                  | Multi          | 64.68                  | 87.83                 |
> |                  | Co             | 98.98                  | 94.50                   |
> | TCL              | Uni            | 96.58                  | 94.45                   |
> |                  | Multi          | 42.19                  | 53.59                   |
> |                  | Co             | 96.52                  | 94.42                   |
>
> Table 1:  Effect of k-NN adaptive Attacks in GAD-VLP for Image-Retrieval task with Flickr30k.
>
> | Model            | Attack          | Non-adaptive | Adaptive |
> |-------------------|-----------------|-------------------------|--------------------------|
> | CLIP$\_{\text{CNN}}\$ | Uni            | 99.67                  | 97.37                   |
> |                  | Co             | 99.59                  | 97.43                  |
> | CLIP$\_{\text{ViT}}\$ | Uni            | 99.92                 | 93.29                  |
> |                  | Co             | 99.01                  | 93.90                  |
> | ALBEF            | Uni            | 94.26                 | 73.85                   |
> |                  | Multi          | 77.39                  | 73.66                 |
> |                  | Co             | 94.27                 | 73.85                   |
> | TCL              | Uni            | 96.02                 | 76.90                   |
> |                  | Multi          | 85.82                  | 78.19                   |
> |                  | Co             | 96.11                  | 78.28                   |
>
> Table 2:  Effect of LID adaptive Attacks in GAD-VLP for Image-Retrieval task with Flickr30k.

---

> ### Author Response · Authors · 2024-11-29
> **Response to Reviewer fo3Q (2/2)**
>
> **A4:**
>
> The attacks proposed after Co-Attack primarily focused on improving transferability. Additionally, both referenced papers utilize PGD-based attacks. As noted in [1]: "Unless specified, during adversarial training, we generate $L_{\infty} = 1/255$ bounded attacks using a 2-step PGD attack with step size $\alpha = 1/255$." Furthermore, Tables 2 and 3 in [2] confirm the use of PGD attacks.
>
> [1] Mao, Chengzhi, et al. "Understanding zero-shot adversarial robustness for large-scale models." arXiv preprint arXiv:2212.07016 (2022).
>
> [2] Schlarmann, Christian, and Matthias Hein. "On the adversarial robustness of multi-modal foundation models." Proceedings of the IEEE/CVF International Conference on Computer Vision. 2023.
>
> ---
>
> **A5:**
>
>  We believe the reviewer's concern is related to 3, where the attacker adaptively makes the adversarial embedding close to clean image embedding to evade the binary detector with LID. We have addressed this in point A3.

---

### Official Review · Reviewer_S3Ts · 2024-11-02

**Soundness:** 2
**Presentation:** 3
**Contribution:** 2
**Rating:** 5
**Confidence:** 4

**Summary:**

This paper proposes GAD-VLP, a geometric adversarial detection framework for vision-language pre-trained models. The method leverages geometric approaches including local intrinsic dimensionality, k-nearest neighbors distance, Mahalanobis distance, and kernel density estimation to identify adversarial examples in VLPs.

**Strengths:**

This paper is easy to follow, and the structure is well-claimed.

**Weaknesses:**

1. The paper lacks rigorous theoretical analysis of why geometric methods work better for VLPs compared to traditional models. While empirical results are shown, there's no formal proof or theoretical guarantees about the method's effectiveness, especially claiming why their approach is fundamentally sound for VLPs.

2. The paper doesn't adequately address the computational costs of calculating geometric metrics, especially for large-scale deployments. Computing k-NN, Mahalanobis distances, and KDE for high-dimensional embeddings can be computationally expensive.

3. The paper doesn't consider sophisticated adaptive attacks that might specifically target the geometric detection mechanism. Adversarial methods often adapt to known defense mechanisms, and the lack of evaluation against such adaptive attacks raises questions about the method's robustness in real-world scenarios.

4. The authors don't thoroughly examine the false positive rates and their impact on model usability. In real-world applications, high false positive rates could lead to unnecessary rejection of legitimate inputs.

**Questions:**

This paper should consider further feedback regarding the lack of rigorous mathematical proof, computational scalability concerns, and vulnerability to adaptive attacks.

---

> ### Author Response · Authors · 2024-11-29
> **Response to Reviewer S3Ts (1/2)**
>
> Thank you for taking the time to review our paper and providing valuable feedback. Below, you will find our responses to your questions:
>
> **A1:**
>
> The theoretical analysis for LID, KNN, and similar methods [1-4] has demonstrated the detectability of adversarial examples under specific assumptions. In this work, we provide evidence that these assumptions also hold for adversarial examples against VLPs. We believe this contribution is valuable, as it extends and builds upon the existing theoretical framework.
>
> [1] Ma, Xingjun, et al. "Characterizing Adversarial Subspaces Using Local Intrinsic Dimensionality." International Conference on Learning Representations. 2018.
>
> [2] Cohen, Gilad, Guillermo Sapiro, and Raja Giryes. "Detecting adversarial samples using influence functions and nearest neighbors." Proceedings of the IEEE/CVF conference on computer vision and pattern recognition. 2020.
>
> [3] Feinman, Reuben, et al. "Detecting adversarial samples from artifacts." arXiv preprint arXiv:1703.00410 (2017).
>
> [4] Lee, Kimin, et al. "A simple unified framework for detecting out-of-distribution samples and adversarial attacks." Advances in neural information processing systems 31 (2018).
>
> ---
>
> **A2:**
>
>
> Thanks for your point. Please find the time cost on the CLIP model (using the RN50 encoder) for each method with the CIFAR-10 dataset using an NVIDIA H100 GPU in the table below (based on seconds). We believe analyzing each query within a range of 0.33 to 5.46 seconds is efficient and practical.
> Also,our framework is significantly more efficient compared to re-training or fine-tuning CLIP for robustness. For example, linear-probe CLIP [1] takes approximately 13 minutes, CoOp [2] requires 14 hours and 40 minutes, and CLIP-Adapter [3] takes about 50 minutes on a n a single NVIDIA GeForce RTX 3090 GPU [4] .
>
>
> | Method       | LID    | $k$-NN | Mahalanobis | KDE    | MCM    |
> |--------------|--------|--------|-------------|--------|--------|
> | **Score**    | 546.11 | 103.19 | 33.08       | 69.82  | 98.48  |
>
> Table 1: Computational cost (seconds) for different methods of GAD-VLP framework
> on CIFAR-10 with CLIPCNN
>
> [1] Radford, Alec, et al. "Learning transferable visual models from natural language supervision." International conference on machine learning. PMLR, 2021.
>
> [2] Zhou, Kaiyang, et al. "Learning to prompt for vision-language models." International Journal of Computer Vision 130.9 (2022): 2337-2348.
>
> [3] Gao, P., Geng, S., Zhang, R., Ma, T., Fang, R., Zhang, Y., Li, H., Qiao, Y.: Clip-adapter: Better vision-language models with feature adapters. arXiv preprint arXiv:2110.04544 (2021).
>
> [4] Zhang, Renrui, et al. "Tip-adapter: Training-free adaption of clip for few-shot classification." European conference on computer vision. Cham: Springer Nature Switzerland, 2022.

---

> ### Author Response · Authors · 2024-11-29
> **Response to Reviewer S3Ts (2/2)**
>
> **A3:**
>
> Thank you for your helpful suggestion. To address this, we have expanded our experimental setup to include adaptive attacks targeting the LID and $k$-NN detection methods. Specifically, we generate attacks designed to optimize for bypassing the detection metrics (LID and k-NN) as follows:
>
> $L_{adaptive}(Z,Z^{'}) = L_{main}(Z,Z^{'}) + \alpha * S(Z^{'})$
>
> Where $S(Z^{'})$ is the function LID or $k$-NN function that computes the score for adversarial embeddings $Z^{'}$ with reference to the clean embeddings $Z$, we put $\alpha= 0.5$ in our experiments. KNN  and LID are both robust when detecting adaptive attacks for CLIP and reasonably effective for ALBEF and TCL. The increase in detection rates against adaptive k-NN attacks can be explained by the dynamics of attack generation. Adaptive attacks targeting the k-NN-based defense incorporate constraints that optimize perturbations around the k-NN structure. The incorporation of the multimodal encoder during attack generation modifies the data distribution, increasing the distinguishability of perturbed samples. This could cause perturbed samples to shift more significantly in the feature space, making them easier to detect. The results are presented in the table below, with additional datasets detailed in subtables a and b of Table 6 in the updated draft (Section A.5 in the Appendix).
>
> | Model            | Attack          | Non-adaptive | Adaptive |
> |-------------------|-----------------|-------------------------|--------------------------|
> | CLIP$\_{\text{CNN}}\$ | Uni            | 99.99                  | 99.99                   |
> |                  | Co             | 99.95                  | 100                  |
> | CLIP$\_{\text{ViT}}\$ | Uni            | 100                 | 100                  |
> |                  | Co             | 99.73                  | 100                  |
> | ALBEF            | Uni            | 99.00                  | 94.57                   |
> |                  | Multi          | 64.68                  | 87.83                 |
> |                  | Co             | 98.98                  | 94.50                   |
> | TCL              | Uni            | 96.58                  | 94.45                   |
> |                  | Multi          | 42.19                  | 53.59                   |
> |                  | Co             | 96.52                  | 94.42                   |
>
> Table 2:  Effect of k-NN adaptive Attacks in GAD-VLP for Image-Retrieval task with Flickr30k.
>
> | Model            | Attack          | Non-adaptive | Adaptive |
> |-------------------|-----------------|-------------------------|--------------------------|
> | CLIP$\_{\text{CNN}}\$ | Uni            | 99.67                  | 97.37                   |
> |                  | Co             | 99.59                  | 97.43                  |
> | CLIP$\_{\text{ViT}}\$ | Uni            | 99.92                 | 93.29                  |
> |                  | Co             | 99.01                  | 93.90                  |
> | ALBEF            | Uni            | 94.26                 | 73.85                   |
> |                  | Multi          | 77.39                  | 73.66                 |
> |                  | Co             | 94.27                 | 73.85                   |
> | TCL              | Uni            | 96.02                 | 76.90                   |
> |                  | Multi          | 85.82                  | 78.19                   |
> |                  | Co             | 96.11                  | 78.28                   |
>
> Table 3:  Effect of LID adaptive Attacks in GAD-VLP for Image-Retrieval task with Flickr30k.
>
> ---
>
> **A4:**
>
> In practice, the defender may adjust the threshold to determine adversarial vs. clean-based applications. This is a classical trade-off for adversarial detection. To ensure the best possible adversarial detection, defenders have to set harsh limits on the threshold that will lead to an increase in FPR. The defender could also reduce unnecessary rejection by relaxing the threshold, which could lead to a decrease in TPR. This threshold value can be determined based on application.
>
> We report AUROC as the main metric so that the trade-off does not impact it. AUROC indicates a higher probability that the score for an adversarial example is higher than the clean one. This metric is independent of the threshold set by a defender. It can fairly assess the detection performance regardless of the threshold. As such, we believe focusing on AUROC is more important than selecting a threshold and then reporting optimized TPR/FPR.
>
> The reported FPR95 corresponds to a 95% TPR, and FPR can be reduced by fine-tuning detection thresholds. However, reducing FPR shouldn’t come at the cost of lower adversarial detection rates. In critical fields like healthcare or autonomous systems, prioritizing a high TPR is essential, as the cost of allowing adversarial examples is much higher than rejecting valid inputs.

---

### Official Review · Reviewer_5u2q · 2024-11-04

**Soundness:** 2
**Presentation:** 3
**Contribution:** 2
**Rating:** 3
**Confidence:** 5

**Summary:**

The paper benchmarks the effectiveness of several methods for detecting adversarial input images in multi-modal models. In particular, these techniques exploit the geometry of the features extracted by the vision and multi-modal encoders to distinguish between clean and adversarial images, and are agnostic of the downstream task. In the experimental evaluation, the different methods are compared on several datasets for both classification and retrieval tasks.

**Strengths:**

- The paper extends the use of several detection techniques to vision-language models, which might be interesting as making these models robust is a relevant challenge. The question of whether detection in multi-modal models is easier than with classifier is also interesting.

- The experimental evaluation includes several datasets and tasks, and different architectures.

**Weaknesses:**

- Detection methods for image classification have often been shown ineffective against adaptive attacks: for example, LID was bypassed in [A], Mahalanobis distance was shown non-robust to adversarial perturbations in [B], and several detection methods are bypassed in [C]. Thus, the proposed methods should be tested against attacks targeting the detection mechanism, e.g. as discussed in [C]. Moreover, the ablation study in Sec. 5.3 about generalization to different attacks only uses methods which are very close to the PGD attack (with only 10 steps) used for tuning the detection methods, and optimize the same loss function: then the generalization to them is not surprising.

- The techniques used for detection are from prior works, which limits the technical novelty of the paper.

- I think it'd be useful to report the success rate of the attack before detection.

[A] https://arxiv.org/abs/1802.00420 \
[B] https://arxiv.org/abs/2201.07012 \
[C] https://openreview.net/forum?id=af1eUDdUVz

**Questions:**

I think it'd be important to test adaptive attacks, especially considering that the same techniques used in this paper have been shown ineffective with standard classifiers.

---

> ### Author Response · Authors · 2024-11-29
> **Response to Reviewer 5u2q (1/2)**
>
> Thank you for taking the time to review our paper and providing valuable feedback. Below, you will find our responses to your questions:
>
> **A1:**
>
> Thank you for your helpful suggestion. To address this, we have expanded our experimental setup to include adaptive attacks targeting the LID and $k$-NN detection methods. Specifically, we generate attacks designed to optimize for bypassing the detection metrics (LID and k-NN) as follows:
>
> $L_{adaptive}(Z,Z^{'}) = L_{main}(Z,Z^{'}) + \alpha * S(Z^{'})$
>
> Where $S(Z^{'})$ is the function LID or $k$-NN function that computes the score for adversarial embeddings $Z^{'}$ with reference to the clean embeddings $Z$, we put $\alpha= 0.5$ in our experiments. KNN  and LID are both robust when detecting adaptive attacks for CLIP and reasonably effective for ALBEF and TCL. The increase in detection rates against adaptive k-NN attacks can be explained by the dynamics of attack generation. Adaptive attacks targeting the k-NN-based defense incorporate constraints that optimize perturbations around the k-NN structure. The incorporation of the multimodal encoder during attack generation modifies the data distribution, increasing the distinguishability of perturbed samples. This could cause perturbed samples to shift more significantly in the feature space, making them easier to detect. The results are presented in the table below, with additional datasets detailed in subtables a and b of Table 6 in the updated draft (Section A.5 in the Appendix).
>
> | Model            | Attack          | Non-adaptive | Adaptive |
> |-------------------|-----------------|-------------------------|--------------------------|
> | CLIP$\_{\text{CNN}}\$ | Uni            | 99.99                  | 99.99                   |
> |                  | Co             | 99.95                  | 100                  |
> | CLIP$\_{\text{ViT}}\$ | Uni            | 100                 | 100                  |
> |                  | Co             | 99.73                  | 100                  |
> | ALBEF            | Uni            | 99.00                  | 94.57                   |
> |                  | Multi          | 64.68                  | 87.83                 |
> |                  | Co             | 98.98                  | 94.50                   |
> | TCL              | Uni            | 96.58                  | 94.45                   |
> |                  | Multi          | 42.19                  | 53.59                   |
> |                  | Co             | 96.52                  | 94.42                   |
>
> Table 1:  Effect of k-NN adaptive Attacks in GAD-VLP for Image-Retrieval task with Flickr30k.
>
> | Model            | Attack          | Non-adaptive | Adaptive |
> |-------------------|-----------------|-------------------------|--------------------------|
> | CLIP$\_{\text{CNN}}\$ | Uni            | 99.67                  | 97.37                   |
> |                  | Co             | 99.59                  | 97.43                  |
> | CLIP$\_{\text{ViT}}\$ | Uni            | 99.92                 | 93.29                  |
> |                  | Co             | 99.01                  | 93.90                  |
> | ALBEF            | Uni            | 94.26                 | 73.85                   |
> |                  | Multi          | 77.39                  | 73.66                 |
> |                  | Co             | 94.27                 | 73.85                   |
> | TCL              | Uni            | 96.02                 | 76.90                   |
> |                  | Multi          | 85.82                  | 78.19                   |
> |                  | Co             | 96.11                  | 78.28                   |
>
> Table 2:  Effect of LID adaptive Attacks in GAD-VLP for Image-Retrieval task with Flickr30k.
>
> ---
>
> **A2:**
>
> > The techniques used for detection are from prior works, which limits the technical novelty of the paper.
>
> To the best of our knowledge, this is the first study to evaluate the detectability of adversarial images for VLPs, addressing a notable gap in the existing literature.
> We believe that demonstrating effective detectability using existing methods in the context of a new problem provides valuable insights to the community. Our rigorous experiments validate this novel finding, which we believe holds equal significance to proposing a new technique.

---

> ### Author Response · Authors · 2024-11-29
> **Response to Reviewer 5u2q (2/2)**
>
> **A3:**
>
> Thank you for your suggestion. We evaluated the attack success rates specifically for image-retrieval tasks across four models and two widely-used datasets. The results, based on IR@1, are provided in the table below. Additional types of attacks and results for both IR@1 and IR@5 can be found in Tables 7 and 8 of the updated draft (Section A.6 in the Appendix).
>
> | Model               | Attack          | Flickr30k | COCO |
> |----------------------|-----------------|--------------|----------|
> | CLIP$\_{\text{CNN}}$ | Uni            | 98.61        | 98.87        |
> |                      | Co             | 99.72        | 99.83        |
> | CLIP$\_{\text{ViT}}$ | Uni            | 97.33        | 98.14        |
> |                      | Co             | 99.42        | 99.21        |
> | ALBEF               | Uni            | 89.50        | 89.18        |
> |                      | Co             | 93.33        | 92.35        |
> | TCL                 | Uni            | 96.18        | 97.56        |
> |                      | Co             | 97.21        | 98.33        |
>
>
> Table 3:  ASR (IR@1) for the Image-Retrieval Task with Flickr30k in aligned VLPs and fused VLPs .

---

> > ### Comment · Reviewer_5u2q · 2024-12-01
> >
> > I thank the authors for the response and additional experiments.
> >
> > However, the adaptive attack loss added in the rebuttal uses a penalty term for the detection score, which is an approach that didn't succeed to bypass LID in [A], while the technique which worked in [A] should be a natural choice to test. Moreover, the orthogonal PGD optimization algorithm from [C], designed to target detection methods, is not used. Thus, I think the evaluation is still not sufficient. Considering this and the limited technical novelty, I will keep my initial score.

---

> ### Author Response · Authors · 2024-12-04
> **Response to Reviewer 5u2q**
>
> We thank the reviewer for evaluating the revised version and for their valuable feedback.
>
> Regarding the statement in paper [A]:
>
> > "Computing the LID term involves determining the $k$-nearest neighbors when calculating $r_i(x)$. Minimizing the gradient of the distance to the current $k$-nearest neighbors does not accurately represent the optimal direction for adjusting the $k$-nearest neighbors."
>
> To address the challenges associated with normal adaptive attacks, we adopted an alternative approach to demonstrate the robustness of our framework. Specifically, we used distinct batches of samples: one for detection and another for generating adversarial points. For instance, in the calculation of $S(Z')$, we employed a batch different from that used for $S(Z)$ during detection. The table below, which shows the AUC detection rate for the TCL model and the adaptive uni-attack, further illustrates the robustness achieved by our framework in this context.
>
> |   | $k$-NN   | LID   |
> |---|-----|-----|
> | Flickr30k | 89.93   | 83.62   |
> | COCO | 73.97   | 80.77   |

---

### Meta-Review · Area_Chair_FH6v · 2024-12-18

**Metareview:**

This work proposes GAD-VLP, Geometric Adversarial Detection for vision-language pre-trained models. The method explores the geometry of the joint vision-language embedding space to detect adversarial examples. However, the reviewers raised several significant concerns.

First, the evaluation of adaptive adversarial attacks, which are stronger and provide a more reliable assessment of the method's effectiveness, was deemed insufficient. Although the authors provided additional results during the rebuttal, these were not sufficient to fully address the reviewers' concerns.

Additionally, there were concerns regarding the lack of theoretical analysis and the potential additional computational costs associated with the proposed method. All reviewers agree that the current work still requires further improvement in these areas.

Therefore, we have decided not to accept this submission.

**Additional Comments On Reviewer Discussion:**

All reviewers agree that the current work still requires further improvement in order to be accepted.

---

### Decision · Program_Chairs · 2025-01-22

Reject